# Object based Scene Representations using Fisher Scores of Local Subspace Projections

**Mandar Dixit and Nuno Vasconcelos**
Department of Electrical and Computer Engineering
University of California, San Diego
{mdixit, nvasconcelos}@ucsd.edu

## Abstract

Several works have shown that deep CNNs can be easily transferred across datasets, e.g. the transfer from object recognition on ImageNet to object detection on Pascal VOC. Less clear, however, is the ability of CNNs to transfer knowledge *across* tasks. A common example of such transfer is the problem of scene classification, that should leverage localized object detections to recognize holistic visual concepts. While this problems is currently addressed with Fisher vector representations, these are now shown ineffective for the high-dimensional and highly non-linear features extracted by modern CNNs. It is argued that this is mostly due to the reliance on a model, the Gaussian mixture of diagonal covariances, which has a very limited ability to capture the second order statistics of CNN features. This problem is addressed by the adoption of a better model, the mixture of factor analyzers (MFA), which approximates the non-linear data manifold by a collection of local sub-spaces. The Fisher score with respect to the MFA (MFA-FS) is derived and proposed as an image representation for holistic image classifiers. Extensive experiments show that the MFA-FS has state of the art performance for object-to-scene transfer and this transfer actually *outperforms* the training of a scene CNN from a large scene dataset. The two representations are also shown to be *complementary,* in the sense that their combination outperforms each of the representations by itself. When combined, they produce a state-of-the-art scene classifier.

## 1   Introduction

In recent years, convolutional neural networks (CNNs) trained on large scale datasets have achieved remarkable performance on traditional vision problems such as image classification [8, 18, 26], object detection and localization [5, 16] and others. The success of CNNs can be attributed to their ability to learn highly discriminative, non-linear, visual transformations with the help of supervised back-propagation [9]. Beyond the impressive, sometimes even superhuman, results on certain datasets, a remarkable property of these classifiers is the solution of the dataset bias problem [20] that has plagued computer vision for decades. It has now been shown many times that a network trained to solve a task on a certain dataset (e.g. object recognition on ImageNet) can be very easily fine-tuned to solve a related problem on another dataset (e.g. object detection on the Pascal VOC or MS-COCO). Less clear, however, is the robustness of current CNNs to the problem of *task bias*, i.e. their ability to generalize *accross tasks*. Given the large number of possible vision tasks, it is impossible to train a CNN from scratch for each. In fact, it is likely not even feasible to collect the large number of images needed to train effective deep CNNs for every task. Hence, there is a need to investigate the problem of task transfer.

In this work, we consider a very common class of such problems, where a classifier trained on a class of instances is to be transferred to a second class of instances, which are *loose combinations* of the

original ones. In particular, we consider the problem where the original instances are *objects* and the target instances are *scene-level concepts* that somehow depend on those objects. Examples of this problem include the transfer of object classifiers to tasks such as scene classification [6, 11, 2] or image captioning [23]. In all these cases, the goal is to predict *holistic scene tags* from the scores (or features) from an object CNN classifier. The dependence of the holistic descriptions on these objects could range from very explicit to very subtle. For example, on the explicit end of the spectrum, an image captioning system could produce sentence such as "a person is sitting on a stool and feeding a zebra." On the other hand, on the subtle end of the spectrum, a scene classification system would leverage the recognition of certain rocks, tree stumps, bushes and a particular lizard species to label an image with the tag "Joshua Tree National Park". While it is obviously possible 1) to collect a large dataset of images, and 2) use them to train a CNN to directly solve each of these tasks, this approach has two main limitations. First, it is extremely time consuming. Second, the "directly learned" CNN will typically not accommodate explicit relations between the holistic descriptions and the objects in the scene. This has, for example, been documented in the scene classification literature, where the performance of the best "directly learned" CNNs [26], can be substantially improved by fusion with object recognition CNNs [6, 11, 2].

So far, the transfer from object CNNs to holistic scene description has been most extensively studied in the area of scene classification, where state of the art results have been obtained with the *bag of semantics* representation of [2]. This consists of feeding image patches through an object recognition CNN, collecting a bag vectors of object recognition scores, and embedding this bag into a fixed dimensional vector space with recourse to a *Fisher vector* [7]. While there are variations of detail, all other competitive methods are based on a similar architecture [6, 11]. This observation is, in principle, applicable to other tasks. For example, the state of the art in image captioning is to use a CNN as an *image encoder* that extracts a feature vector from the image. This feature vector is the fed to a natural language decoder (typically an LSTM) that produces sentences. While there has not yet been an extensive investigation of the best image encoder, it is likely that the best representations for scene classification should also be effective encodings for language generation. For these reasons, we restrict our attention to the scene classifcation problem in the remainder of this work, focusing on the question of how to address possible limitations of the Fisher vector embedding. We note, in particular, that while Fisher vectors have been classically defined using gradients of image log-likelihood with respect to the means and variances of a Gaussian mixture model (GMM) [13], this definition has not been applied universally in the CNN transfer context, where variance statistics are often disregarded [6, 2].

In this work we make several contributions to the use of Fisher vector type of representations for object to scene transfer. The first is to show that, for object recognition scores produced by a CNN [2], variance statistics are much less informative of scene class distributions than the mean gradients, and can even degrade scene classification performance. We then argue that this is due to the inability of the standard GMM of diagonal covariances to provide a good approximation to the non-linear manifold of CNN responses. This leads to the adoption of a richer generative model, the mixture of factor analyzers (MFA) [4, 22], which locally approximates the scene class manifold by low-dimensional linear spaces. Our second contribution is to show that, by locally projecting the feature data into these spaces, the MFA can efficiently model its local covariance structure. For this, we derive the Fisher score of the MFA model, denoted the MFA Fisher score (MFA-FS), a representation similar to the GMM Fisher vector of [13, 17]. We show that, for high dimensional CNN features, the MFA-FS captures highly discriminative covariance statistics, which were previously unavailable in [6, 2], producing significantly improved scene classification over the conventional GMM Fisher vector. The third contribution is a detailed experimental investigation of the MFA-FS. Since this can be seen as a second order pooling mechanism, we compare it to a number of recent methods for second order pooling of CNN features [21, 3]. Although these methods describe global covariance structure, they lack the ability of the MFA-FS to capture that information along locally linear approximations of the highly non-linear CNN feature manifold. This is shown to be important, as the MFA-FS is shown to outperform all these representations by non-trivial margins. Finally, we show that the MFA-FS enables effective task transfer, by showing that MFA-FS vectors extracted from deep CNNs trained for ImageNet object recognition [8, 18], achieve state-of-the-art results on challenging scene recognition benchmarks, such as SUN [25] and MIT Indoor Scenes [14].

## 2  Fisher scores

In computer vision, an image $\mathcal{I}$ is frequently interpreted as a set of descriptors $\mathcal{D} = \{x_1, \ldots, x_n\}$ sampled from some generative model $p(x; \theta)$. Since most classifiers require fixed-length inputs, it is common to map the set $\mathcal{D}$ into a fixed-length vector. A popular mapping consists of computing the gradient (with respect to $\theta$) of the log-likelihood $\nabla_\theta L(\theta) = \frac{\partial}{\partial \theta} \log p(\mathcal{D}; \theta)$ for a model $\theta^b$. This is known as the *Fisher score* of $\theta$. This gradient vector is often normalized by the square root of the Fisher information matrix $\mathcal{F}$, according to $\mathcal{F}^{-\frac{1}{2}} \nabla_\theta L(\theta)$. This is referred to as the Fisher vector (FV) [7] representation of $\mathcal{I}$. While the Fisher vector is frequently used with a Gaussian mixture model (GMM) [13, 17], any generative model $p(x; \theta)$ can be used. However, the information matrix is not always easy to compute. When this is case, it is common to rely on the simpler representation of $\mathcal{I}$ by the score $\nabla_\theta L(\theta)$. This is, for example, the case with the sparse coded gradient vectors in [11]. We next show that, for models with hidden variables, the Fisher score can be obtained trivially from the steps of the expectation maximization (EM) algorithm commonly used to learn such models.

### 2.1  Fisher Scores from EM

Consider the log-likelihood of $\mathcal{D}$ under a latent-variable model $\log p(\mathcal{D}; \theta) = \log \int p(\mathcal{D}, z; \theta) dz$ of hidden variable $z$. Since the left-hand side does not depend on the hidden variable, this can be written in an alternate form, which is widely used in the EM literature,

$$
\begin{aligned}
\log p(\mathcal{D}; \theta) &= \int q(z) \log p(\mathcal{D}, z; \theta) dz - \int q(z) \log q(z) dz + \int q(z) \log \frac{q(z)}{p(z|\mathcal{D}; \theta)} dz \\
&= Q(q; \theta) + H(q) + KL(q\|p; \theta)
\end{aligned}
\tag{1}
$$

where $Q(q; \theta)$ is the "Q" function, $q(z)$ a general probability distribution, $H(q)$ its differential entropy and $KL(q\|p; \theta)$ the Kullback Liebler divergence between the posterior $p(z|\mathcal{D}; \theta)$ and $q(z)$. Hence,

$$
\frac{\partial}{\partial \theta} \log p(\mathcal{D}; \theta) = \frac{\partial}{\partial \theta} Q(q; \theta) + \frac{\partial}{\partial \theta} KL(q\|p; \theta)
\tag{2}
$$

where

$$
\frac{\partial}{\partial \theta} KL(q\|p; \theta) = -\int \frac{q(z)}{p(z|\mathcal{D}; \theta)} \frac{\partial}{\partial \theta} p(z|\mathcal{D}; \theta) dz.
\tag{3}
$$

In each iteration of the EM algorithm the $q$ distribution is chosen as $q(z) = p(z|\mathcal{D}; \theta^b)$, where $\theta^b$ is a reference parameter vector (the parameter estimates from the previous EM iteration) and

$$
Q(q; \theta) = \int p(z|\mathcal{D}; \theta^b) \log p(\mathcal{D}, z; \theta) dz = E_{z|\mathcal{D};\theta^b}[\log p(\mathcal{D}, z; \theta)].
\tag{4}
$$

It follows that

$$
\frac{\partial}{\partial \theta} KL(q\|p; \theta)\Big|_{\theta=\theta^b} = -\int \frac{p(z|\mathcal{D}; \theta^b)}{p(z|\mathcal{D}; \theta^b)} \frac{\partial}{\partial \theta} p(z|\mathcal{D}; \theta)\Big|_{\theta=\theta^b} dz = -\frac{\partial}{\partial \theta} \int p(z|\mathcal{D}; \theta)\Big|_{\theta=\theta^b} dz = 0
$$

and

$$
\frac{\partial}{\partial \theta} \log p(\mathcal{D}; \theta)\Big|_{\theta=\theta^b} = \frac{\partial}{\partial \theta} Q(p(z|\mathcal{D}; \theta^b); \theta)\Big|_{\theta=\theta^b}.
\tag{5}
$$

In summary, the Fisher score $\nabla_\theta L(\theta)|_{\{\theta=\theta^b\}}$ of background model $\theta^b$ is the gradient of the Q-function of EM evaluated at reference model $\theta^b$. The computation of the score thus simplifies into the two steps of EM. First, the E step computes the Q function $Q(p(z|x; \theta^b); \theta)$ at the reference $\theta^b$. Second, the M-step evaluates the gradient of the Q function with respect to $\theta$ at $\theta = \theta^b$. This interpretation of the Fisher score is particularly helpful when efficient implementations of the EM algorithm are available, e.g. the recursive Baum-Welch computations commonly used to learn hidden Markov models [15]. For more tractable distributions, such as the GMM, it enables the simple reuse of the EM equations, which are always required to learn the reference model $\theta^b$, to compute the Fisher score.

## 2.2 Bag of features

Fisher scores are usually combined with the bag-of-features representation, where an image is described as an orderless collection of localized descriptors $\mathcal{D} = \{x_1, x_2, \ldots x_n\}$. These were traditionally SIFT descriptors, but have more recently been replaced with responses of object recognition CNNs [6, 1, 2]. In this work we use the semantic features proposed in [2], which are obtained by transforming softmax probability vectors $p_i$, obtained for image patches, into their natural parameter form. These features were shown to perform better than activations of other CNN layers [2].

## 2.3 Gaussian Mixture Fisher Vectors

A GMM is a model with a discrete hidden variable that determines the mixture component which explains the observed data. The generative process is as follows. A mixture component $z_i$ is first sampled from a multinomial distribution $p(z = k) = w_k$. An observation $x_i$ is then sampled from the Gaussian component $p(x|z = k) \sim \mathcal{G}(x, \mu_k, \sigma_k)$ of mean $\mu_k$ and variance $\sigma_k$. Both the hidden and observed variables are sampled independently, and the Q function simplifies to

$$
\begin{aligned}
Q(p(z|\mathcal{D}; \theta^b); \theta) &= \sum_i E_{z_i|x_i; \theta^b} \left[ \sum_k I(z_i, k) \log p(x_i, k; \theta) \right] \\
&= \sum_{i,k} h_{ik} \log p(x_i|z_i = k; \theta) w_k
\end{aligned}
\tag{6}
$$

where $I(.)$ is the indicator function and $h_{ik}$ is the posterior probability $p(k|x_i; \theta^b)$. The probability vectors $h_i$ are the only quantities computed in the E-step.

In the Fisher vector literature [13, 17], the GMM is assumed to have diagonal covariances. This is denoted as the variance-GMM. Substituting the expressions of $p(x_i|z_i = k; \theta)$ and differentiating the Q function with respect to parameters $\theta = \{\mu_k, \sigma_k\}$ leads to the two components of the Fisher score

$$
\mathcal{G}_{\mu_k^d}(\mathcal{I}) = \frac{\partial}{\partial \mu_k^d} L(\theta) = \sum_i p(k|x_i) \left( \frac{x_i^d - \mu_k^d}{(\sigma_k^d)^2} \right)
\tag{7}
$$

$$
\mathcal{G}_{\sigma_k^d}(\mathcal{I}) = \frac{\partial}{\partial \sigma_k^d} L(\theta) = \sum_i p(k|x_i) \left[ \frac{(x_i^d - \mu_k^d)^2}{(\sigma_k^d)^3} - \frac{1}{\sigma_k^d} \right].
\tag{8}
$$

These quantities are evaluated using a reference model $\theta^b = \{\mu_k^b, \sigma_k^b\}$ learned (with EM) from all training data. To compute the Fisher vectors, scores in (7) and (8) are often scaled by an approximate Fisher information matrix, as detailed in [17]. When used with SIFT descriptors, these mean and variance scores usually capture complimentary discriminative information, useful for image classification [13]. Yet, FVs computed from CNN features only use the mean gradients similar to (7), ignoring second-order statistics [6, 2]. In the experimental section, we show that the variance statistics of CNN features perform poorly compared to the mean gradients. This is perhaps due to the inability of the variance-GMM to accurately model data in high dimensions. We test this hypothesis by considering a model better suited for this task.

## 2.4 Fisher Scores for the Mixture of Factor Analyzers

A factor analyzer (FA) is a type of a Gaussian distribution that models high dimensional observations $x \in \mathbb{R}^D$ in terms of latent variables or "factors" $z \in \mathbb{R}^R$ defined on a low-dimensional subspace $R << D$ [4]. The process can be written as $x = \Lambda z + \epsilon$, where $\Lambda$ is known as the factor loading matrix and $\epsilon$ models the additive noise in dimensions of $x$. Factors $z$ are assumed distributed as $\mathcal{G}(z, 0, I)$ and the noise is assumed to be $\mathcal{G}(\epsilon, 0, \psi)$, where $\psi$ is a diagonal matrix. It can be shown that $x$ has full covariance $S = \Lambda\Lambda^T + \psi$, making the FA better suited for high dimensional modeling than a Gaussian of diagonal covariance.

A mixture of factor analyzers (MFA) is an extension of the FA that allows a piece-wise linear approximation of a non-linear data manifold. Unlike the GMM, it has two hidden variables: a discrete variable $s$, $p(s = k) = w_k$, which determines the mixture assignments and a continuous latent variable $z \in \mathbb{R}^R$, $p(z|s = k) = \mathcal{G}(z, 0, I)$, which is a low dimensional projection of the observation variable $x \in \mathbb{R}^D$, $p(x|z, s = k) = \mathcal{G}(x, \Lambda_k z + \mu_k, \psi)$. Hence, the $k^{th}$ MFA component is a FA of mean $\mu_k$ and subspace defined by $\Lambda_k$. Overall, the MFA components approximate the distribution of

the observations $x$ by a set of sub-spaces in observation space. The Q function is

$$Q(p(s,z|\mathcal{D};\theta^b);\theta) = \sum_i E_{z_i,s_i|x_i;\theta^b}\left[\sum_k I(s_i,k)\log p(x_i,z_i,s_i=k;\theta)\right] \tag{9}$$

$$= \sum_{i,k} h_{ik} E_{z_i|x_i;\theta^b}\left[\log G(x_i,\Lambda_k z_i + \mu_k,\psi) + \log G(z_i,0,I) + \log w_k\right]. \tag{10}$$

where $h_{ik} = p(s_i = k|x_i;\theta^b)$. After some simplifications, the E step reduces to computing

$$h_{ik} = p(k|x_i;\theta^b) \propto w_k^b \mathcal{N}(x_i,\mu_k^b,S_k^b) \tag{11}$$

$$E_{z_i|x_i;\theta^b}[z_i] = \beta_k^b(x_i - \mu_k^b) \tag{12}$$

$$E_{z_i|x_i;\theta^b}[z_i z_i^T] = \left(I - \beta_k^b \Lambda_k^b + \beta_k^b(x_i - \mu_k^b)(x_i - \mu_k^b)^T \beta_k^{b^T}\right) \tag{13}$$

with $S_k^b = \Lambda_k^b \Lambda_k^{b^T} + \psi^b$ and $\beta_k^b = \Lambda_k^{b^T}\left(S_k^b\right)^{-1}$. The M-step then evaluates the Fisher score of $\theta = \{\mu_k^b, \Lambda_k^b\}$. With some algebraic manipulations, this can be shown to have components

$$\mathcal{G}_{\mu_k}(\mathcal{I}) = \sum_i p(k|x_i;\theta^b)\psi^{b^{-1}}\left(I - \Lambda_k^b \beta_k^b\right)(x_i - \mu_k^b) \tag{14}$$

$$\mathcal{G}_{\Lambda_k}(\mathcal{I}) = \sum_i p(k|x_i;\theta^b)\psi^{b^{-1}}(\Lambda_k^b \beta_k^b - I)\left[(x_i - \mu_k^b)(x_i - \mu_k^b)^T \beta_k^{b^T} - \Lambda_k^b\right]. \tag{15}$$

For a detailed discussion of the Q function, the reader is referred to the EM derivation in [4]. Note that the scores with respect to the means are functionally similar to the first order residuals in (7). However, the scores with respect to the factor loading matrices $\Lambda_k$ account for covariance statistics of the observations $x_i$, not just variances. We refer to the representations (14) and (15) as MFA Fisher scores (MFA-FS). Note that these are not FVs due to the absence of normalization by the Fisher information, which is more complex to compute than for the variance-GMM.

## 3 Related work

The most popular approach to transfer object scores (usually from an ImageNet CNN) into a feature vector for scene classification is to rely on FV-style pooling. Although most classifiers default to the GMM-FV embedding [6, 1, 2, 24], some recent works have explored different encoding [11] and pooling schemes [21, 3] with promising results. Liu *et al.* [11] derived an FV like representation from sparse coding. Their model can be described as a factor analyzer with Gaussian observations $p(x|z) \sim \mathcal{N}(\Lambda z, \sigma^2 I)$ conditioned on Laplace factors $p(z) \propto \prod_r \exp(-|z_r|)$. While the sparse FA marginal $p(x)$ is intractable, it can be approximated by an evidence lower bound $p(x) \geq \int q(z)\frac{p(x,z)}{q(z)}dz$ derived from a suitable variational posterior $q(z)$. In [11], $q$ is a point posterior $\delta(z - z^*)$ and the MAP inference simplifies into sparse coding. The image representation is obtained using gradients of the sparse coding objective evaluated at the MAP factors $z^*$, with respect to the factor loadings $\Lambda$. [21] proposed an alternative bilinear pooling mechanism $\sum_i x_i x_i^T$. Similar to the MFA-FS, this captures second order statistics of CNN feature space, albeit globally. Due to its simplicity, this mechanism supports fine-tuning of the object CNN to scene classes. Gao *et al.* [3] have recently shown that this representation can be compressed with minimal performance loss.

## 4 Experiments

The MFA-FS was evaluated on the scene classification problem, using the 67 class MIT Indoor scenes dataset [14] and the 397 class MIT SUN dataset [25]. For Indoor scenes, a single training set of 80 images per class is provided by the authors. The test set consists of 20 images per class. Results are reported as average per class classification accuracy. The authors of SUN provide multiple train/test splits, each test set containing 50 images per class. Results are reported as mean average per class classification accuracy over splits. Three object recognition CNNs, pre-trained on ImageNet, were used to extract features: the 8 layer network of [8] (denoted as AlexNet) and the deeper 16 and 19 layer networks of [18] (denoted VGG-16 and VGG-19, respectively). These CNNs assign 1000 dimensional object recognition probabilities to $P \times P$ patches (sampled on a grid of fixed spacing) of the scene images, with $P \in \{128, 160, 96\}$. Patch probability vectors were converted into their natural parameter form and PCA-reduced to 500 dimensions as in [2]. Each image was mapped into a

Table 1: Classification accuracy ($K = 50$, $R = 10$).

| MIT Indoor | |
|---|---|
| GMM FV ($\mu$) | 66.08 |
| GMM FV ($\sigma$) | 53.86 |
| MFA FS ($\mu$) | 67.68 |
| MFA FS ($\Lambda$) | **71.11** |

| SUN | |
|---|---|
| GMM FV ($\mu$) | 50.01 |
| GMM FV ($\sigma$) | 37.71 |
| MFA FS ($\mu$) | 51.43 |
| MFA FS ($\Lambda$) | **53.38** |

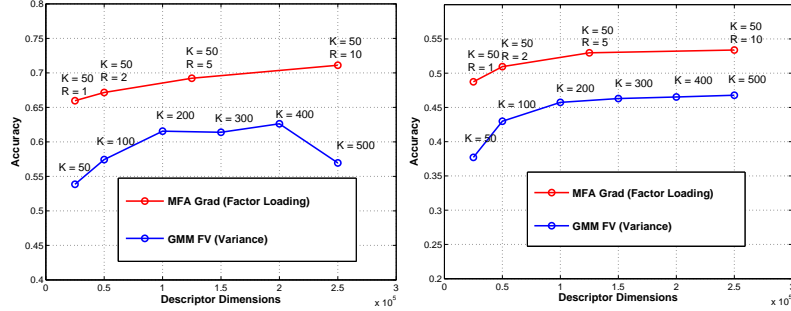

Figure 1: Classification accuracy vs. descriptor size for MFA-FS($\Lambda$) of $K = 50$ components and $R$ factor dimensions and GMM-FV($\sigma$) of $K$ components. Left: MIT Indoor. Right: SUN.

GMM-FV [13] using a background GMM, and an MFA-FS, using (14), (15) and a background MFA. As usual in the FV literature, these vectors were power normalized, $L2$ normalized, and classified with a cross-validated linear SVM. These classifiers were compared to scene CNNs, trained on the large scale Places dataset. In this case, the features from the penultimate CNN layer were used as a holistic scene representation and classified with a linear SVM, as in [26]. We used the places CNNs with the AlexNet and VGG-16 architectures provided by the authors.

## 4.1 Impact of Covariance Modeling

We begin with an experiment to compare the modeling power of MFAs to variance-GMMs. This was based on AlexNet features, mixtures of $K = 50$ components, and an MFA latent space dimension of $R = 10$. Table 1 presents the classification accuracy of a GMM-FV that only considers the mean - GMM-FV($\mu$) - or variance - GMM-FV($\sigma$) - parameters and a MFA-FS that only considers the mean - MFA-FS($\mu$) - or covariance - MFA-FS($\Lambda$) - parameters. The most interesting observation is the complete failure of the GMM-FV ($\sigma$), which under-performs the GMM-FV($\mu$) by more than 10%. The difference between the two components of the GMM-FV is not as startling for lower dimensional SIFT features [13]. However, for CNN features, the discriminative power of variance statistics is exceptionally low. This explains why previous FV representations for CNNs [6, 2] only consider gradients with respect to the means. A second observation of importance is that the improved modeling of covariances by the MFA eliminates this problem. In fact, MFA-FS($\Lambda$) is significantly better than both GMM-FVs. It could be argued that a fair comparison requires an increase in the GMM modeling capacity. Fig. 1 tests this hypothesis by comparing GMM-FVs($\sigma$) and MFA-FS ($\Lambda$) for various numbers of GMM components ($K \in \{50, \ldots, 500\}$) and MFA hidden sub-spaces dimensions ($R \in \{1, \ldots, 10\}$). For comparable vector dimensions, the covariance based scores always significantly outperforms the variance statistics on both datasets. A final observation is that, due to covariance modeling in MFAs, the MFA-FS($\mu$) performs better the GMM-FV($\mu$). The first order residuals pooled to obtain the MFA-FS($\mu$) (14) are scaled by covariance matrices instead of variances. This local de-correlation provides a non-trivial improvement for the MFA-FS($\mu$) over the GMM-FV($\mu$)($\sim 1.5\%$ points). Covariance modeling was previously used in [19] to obtain FVs w.r.t. Gaussian means and local subspace variances (eigen-values of covariance). Their subspace variance FV, derived with our MFAs, performs much better than the variance GMM-FV ($\sigma$), due to a better underlying model ($60.7\%$ v $53.86\%$ on Indoor). It is, however, still inferior to the MFA-FS($\Lambda$) which captures full covariance within local subspaces.

While a combination of the MFA-FS($\mu$) and MFA-FS($\Lambda$) produces a small improvement ($\sim 1\%$), we restrict to using the latter in the remainder of this work.

## 4.2 Multi-scale learning and Deep CNNs

Recent works have demonstrated value in combining deep CNN features extracted at multiple-scales. Table 2 presents the classification accuracies of the MFA-FS ($\Lambda$) based on AlexNet, and 16 and 19 layer VGG features extracted from 96x96, 128x128 and 160x160 pixel image patches, as well as their concatenation (3 scales), as suggested by [2]. These results confirm the benefits of multi-scale feature combination, which achieves the best performance for all CNNs and datasets.

Table 2: MFA-FS classification accuracy as a function of patch scale.

| | MIT Indoor | SUN |
|---|---|---|
| AlexNet | | |
| 160x160 | 69.83 | 52.36 |
| 128x128 | 71.11 | 53.38 |
| 96x96 | 70.51 | 53.54 |
| 3 scales | **73.58** | **55.95** |
| VGG-16 | | |
| 160x160 | 77.26 | 59.77 |
| 128x128 | 77.28 | 60.99 |
| 96x96 | 79.57 | 61.71 |
| 3 scales | **80.1** | **63.31** |
| VGG-19 | | |
| 160x160 | 77.21 | - |
| 128x128 | 79.39 | - |
| 96x96 | 79.9 | - |
| 3 scales | **81.43** | - |

Table 3: Performance of scene classification methods. *-combination of patch scales (128, 96, 160).

| Method | MIT Indoor | SUN |
|---|---|---|
| MFA-FS + Places (VGG) | **87.23** | **71.06** |
| MFA-FS + Places (AlexNet) | 79.86 | 63.16 |
| MFA-FS (VGG) | 81.43 | 63.31 |
| MFA-FS (AlexNet) | 73.58 | 55.95 |
| Full BN (VGG) [3] | 77.55 | - |
| Compact BN (VGG) [3] | 76.17 | - |
| H-Sparse (VGG) [12] | 79.5 | - |
| Sparse Coding (VGG) [12] | 77.6 | - |
| Sparse Coding (AlexNet) [11] | 68.2 | |
| MetaClass (AlexNet) + Places [24] | 78.9 | 58.11 |
| FV (AlexNet)(4 scales) + Places [2] | 79.0 | 61.72 |
| FV (AlexNet)(3 scales) + Places [2] | 78.5* | - |
| FV (AlexNet) (4 scales) [2] | 72.86 | 54.4 |
| FV (Alexnet)(3 scales) [2] | 71.24 | 53.0 |
| VLAD (AlexNet) [6] | 68.88 | 51.98 |
| FV+FC (VGG) [1] | 81.0 | - |
| Mid Level [10] | 70.46 | - |

## 4.3 Comparison with ImageNet based Classifiers

We next compared the MFA-FS to state of the art scene classifiers also based on transfer from ImageNet CNN features [11, 1–3]. Since all these methods only report results for MIT Indoor, we limited the comparison to this dataset, with the results of Table 4. The GMM-FV of [2] operates on AlexNet CNN semantics extracted from image patches of multiple sizes (96, 128, 160, 80). The FV in [1] is computed using convolutional features from AlexNet or VGG-16 extracted in a large multi-scale setting. Liu *et al.* proposed a gradient representation based on sparse codes. Their initial results were reported on a single patch scale of 128x128 using AlexNet features [11]. More recently, they have proposed an improved H-Sparse representation, combined multiple patch scales and used VGG features in [12]. The recently proposed bilinear (BN) descriptor pooling of [21] is similar to the MFA-FS in the sense that it captures global second order descriptor statistics. The simplicity of these descriptors enables the fine-tuning of the CNN layers to the scene classification task. However, their results, reproduced in [3] for VGG-16 features, are clearly inferior to those of the MFA-FS without fine-tuning. Gao *et al.* [3] propose a way to compress these bilinear statistics with trainable transformations. For a compact image representation of size $8K$, their accuracy is inferior to a representation of $5K$ dimensions obtained by combining the MFA-FS with a simple PCA.

These experiments show that the MFA-FS is a state of the art procedure for task transfer from object recognition (on ImageNet) to scene classification (e.g. on MIT Indoor or SUN). Its closest competitor is the classifier of [1], which combines CNN features in a massive multiscale setting ( 10 image sizes). While MFA-FS outperforms [1] with only 3 image scales, its performance improves even further with addition of more scales ($82\%$ with VGG, 4 patch sizes).

## 4.4 Task transfer performance

The next question is how object-to-scene transfer compares to the much more intensive process, pursued by [26], of collecting a large scale labeled dataset and training a deep CNN from it. Their scene dataset, known as *Places*, consists of 2.4M images, from which both AlexNet and VGG Net CNNs were trained for scene classification. The fully connected features from the networks are used as scene representations and classified with linear SVMs on Indoor scenes and SUN. The Places CNN features are a direct alternatives to the MFA-FS. While the use of the former is an example of dataset transfer (features trained on scenes to classify scenes) the use of the latter is an example of task transfer (features trained on objects to classify scenes).

A comparison between the two transfer approaches is shown in table 5. Somewhat surprisingly, task transfer with the MFA-FS *outperformed* dataset transfer with the Places CNN, on both MIT Indoors and SUN and for both the AlexNet and VGG architectures. This supports the hypothesis that the variability of configurations of most scenes makes scene classification much harder than object recognition, to the point where CNN architectures that have close-to or above human performance for

Table 4: Comparison to task transfer methods (ImageNet CNNs) on MIT Indoor.

| Method | 1 scale | mscale |
|---|---|---|
| AlexNet | | |
| MFA-FS | **71.11** | **73.58** |
| GMM FV [2] | 68.5 | 72.86 |
| FV+FC [1] | - | 71.6 |
| Sparse Coding [11] | 68.2 | - |
| VGG | | |
| MFA-FS | **79.9** | **81.43** |
| Sparse Coding [12] | - | 77.6 |
| H-Sparse [12] | - | 79.5 |
| BN [3] | 77.55 | - |
| FV+FC [1] | - | 81.0 |
| VGG + dim. reduction | | |
| MFA-FS + PCA (5k) | **79.3** | - |
| BN (8k) [3] | 76.17 | - |

Table 5: Comparison with the Places trained Scene CNNs.

| Method | SUN | Indoor |
|---|---|---|
| AlexNet | | |
| MFA-FS | 55.95 | 73.58 |
| Places | 54.3 | 68.24 |
| Combined | 63.16 | 79.86 |
| VGG | | |
| MFA-FS | 63.31 | 81.43 |
| Places | 61.32 | 79.47 |
| Combined | **71.06** | **87.23** |
| AlexNet + VGG | | |
| Places (VGG + Alex) | 65.91 | 81.29 |
| MFA-FS(Alex) + Places(VGG) | 68.8 | 85.6 |
| MFA-FS(VGG) + Places(Alex) | 67.34 | 82.82 |

object recognition are much less effective for scenes. It is, instead, preferable to pool object detections across the scene image, using a pooling mechanism such as the MFA-FS. This observation is in line with an interesting result of [2], showing that the object-based and scene-based representations are *complementary,* by concatenating ImageNet- and Places-based feature vectors. By replacing the the GMM-FV of [2] with the MFA-FS now proposed, we improve upon these results. For both the AlexNet and VGG CNNs, the combination of the ImageNet-based MFA-FS and the Places CNN feature vector outperformed both the MFA-FS and the Places CNN features by themselves, in both SUN and MIT Indoor. To the best of our knowledge, no method using these or deeper CNNs has reported better results than the combined MFA-FS and Places VGG features of Table 5.

It could be argued that this improvement is just an effect of the often observed benefits of fusing different classifiers. Many works even resort to "bagging" of multiple CNNs to achieve performance improvements [18]. To test this hypothesis we also implemented a classifier that combines two Places CNNs with the AlexNet and VGG architectures. This is shown as Places (VGG+AlexNet) in the last section of Table 5. While improving on the performance of both MFA-FS and Places, its performance is not as good as that of the combination of the object-based and scene-based representations (MFA-FS + Places). As shown in the remainder of the last section of the table, any combination of an object CNN with MFA-FS based transfer and a scene CNN outperforms this classifier.

Finally, table 3 compares results to the best recent scene classification methods in the literature. This comparison shows that MFA-FS + Places combination is a state-of-the-art classifier with substantial gains over all other proposals. The results of $71.06\%$ on SUN and $87.23\%$ on Indoor scenes substantially outperform the previous best results of $61.7\%$ and $81.7\%$, respectively.

# 5   Conclusion

It is now well established that deep CNNs can be transferred across datasets that address similar tasks. It is less clear, however, whether they are robust to transfer *across* tasks. In this work, we have considered a class of problems that involve this type of transfer, namely problems that benefit from transferring object detections into holistic scene level inference, eg. scene classification. While such problems have been addressed with FV-like representations in the past, we have shown that these are not very effective for the high-dimensional CNN features. The reason is their reliance on a model, the variance-GMM, with a limited flexibility. We have addressed this problem by adopting a better model, the MFA, which approximates the non-linear data manifold by a set of local sub-spaces. We then introduced the Fisher score with respect to this model, denoted as the MFA-FS. Through extensive experiments, we have shown that the MFA-FS has state of the art performance for object-to-scene transfer and this transfer actually *outperforms* a scene CNN trained on a large scene dataset. These results are significant given that 1) MFA training takes only a few hours versus training a CNN, and 2) transfer requires a much smaller scene dataset.

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
