[Reviews · NeurIPS 2016]

Reviewer 1

Summary

The paper addresses the problem of performing scene classification from images. Specifically, it focuses on the scenario where classifiers trained on object classes are used to predict scene classes, which they label a task-transfer procedure (using classifiers for object-level concepts to predict scene-level concepts). The paper gives a formal introduction to Fisher Vectors [7] and Mixture of factor analyzers[4,20], which serve to properly introduce their proposed method. Following this, the paper proposes "MFA-FS" a method which encodes the Fisher score of the mixture of factor analyzers [4,20]. This is followed by an evaluation performed in the MIT 67 (67 classes) and the MIT SUN (397 classes) datasets. The experimental results show that: a) for high-dimensional descriptors, the covariance-based scores always outperform methods based on variance statistics on both dataset. b) the proposed method outperforms state-of-the-art methods (compared against [2,3,10]). c) the proposed method (following the task-transfer paradigm) is able to outperform methods based on CNNs trained with scene-level annotations (dataset-transfer paradigm) when either an AlexNet or VGG architecture is used.

Qualitative Assessment

Strong Points: - I appreciate how the paper the formally introduced all the foundations necessary to understand the proposed MFA-FS method. In this sense the paper feels self-contained. - I do also appreciate the extensive evaluation presented on this paper. The analysis on how different factors (the effect of covariance statistics, multi-scale/depth modeling, task vs. dataset transferability) affect classification performance, and the added discussions, provides an insight of why the proposed methods actually work . Likewise, the comparison against the state-of-the-art methods shows the competitiveness of the proposed approach. Weak Points: Even thought scene classification is a problem with relatively high-level of attention, the related work section (Section 3) is very reduced. Reading about the task-transfer paradigm addressed in this paper, the work from Li et al.,CVPR'15 "Mid-level Deep Pattern Mining" came to my mind. Similar to this, there is a group of works focused on mining mid-level elements(image patches) which address the same problem scene classification based on object-level CNN features. Perhaps, addressing the scene classification literature focused on mining mid-level elements in the Related work section, and comparing the performance accordingly (in Section 4.3) should strengthen the quality of the paper.

Confidence in this Review

1-Less confident (might not have understood significant parts)


Reviewer 2

Summary

The submission describes a novel variant of the Fisher score, with respect to a mixture of factor analyzers, as opposed to the often chosen mixture of Gaussians [with diagonal covariance]. The authors argue that nonlinear data manifolds can be better approximated by this model and present an experimental evaluation on a task transfer problem. The experiments focus on scene classification, where the output from object-level classifiers is embedded as a global representation by means of Fisher score w.r.t. the proposed model, leading to greatly improved results. It is also shown that these object-level features are somewhat complementary to learning a scene classification network from scratch; as a result, the previous state of the art is improved upon.

Qualitative Assessment

I enjoyed reading the well-written submission. The contributions are incrementally building upon previous work, but the novelty of the chosen approach is presented clearly, leading to new state of the art results. Section 2 nicely reiterates the computation of Fisher scores for the GMM, and then presents the computation w.r.t. the mixture of factor analyzers. Despite being math heavy, the section is written in an understandable way, if one additionally consults reference [4]. The experiments are convincing w.r.t. the achieved results. Since the "expressiveness" of the MFA appears to be somewhere in between a GMM with diagonal covariances and a GMM with full covariances, it would be great to have an additional set of experiments comparing against a Fisher score model based on full-cov. GMMs. See Tanaka et al., "Fisher Vector based on Full-covariance Gaussian Mixture Model" (https://www.jstage.jst.go.jp/article/ipsjtcva/5/0/5_50/_article), which should probably be added as a reference. How does the performance for each of these three different generative models (diag-cov-GMM, MFA, full-cov-GMM) change w.r.t. to the amount of training data allocated for learning the respective model? I could imagine that there is a trade-off somwehere, since more parameters need to be estimated for the more complex models. And as a smaller comment: In Section 4.1, how does the performance change for the MFA-FS when the number of components K is changed? The rest of the experimental section is without fault, methodically answering potentially remaining questions (e.g. ll. 281-288) and presenting new state of the art results. Overall, this is a submission of high quality, which deserves to be presented at NIPS and be picked up by the scene classification community as well as beyond (e.g. applications to retrieval).

Confidence in this Review

2-Confident (read it all; understood it all reasonably well)


Reviewer 3

Summary

The paper proposes to use mixture of factor analyzers (MFA) instead of Gaussian Mixture Models (GMM) as the generative model to compute the Fisher Vector (FV). It gives the detailed derivation of MFA based FV. Experimental results show that the proposed method achieves SoA performance for scene classification.

Qualitative Assessment

The paper proposes using MFA to provide an approximation to the non-linear manifold of CNN responses, and compute the FV respect to MFA as image representations. This method achieves better performance than traditional FV for scene classification. But there are some limitations. 1. It seems that the paper just simply uses MFA instead of GMM as the generative model, so the technical novelty is limited. 2. The paper claimed that the GMM of diagonal covariances cannot provide a good approximation to the non-linear manifold of CNN responses, while MFA is better, but do not provide any theoretical proof. 3. The paper should also do some discussions with the GMM based FV method adopted in reference [1]. And actually, when only using the FV, performance in [1] on MITindoor is 81.0, which is only a little worse than the MFA-FS (81.43).

Confidence in this Review

2-Confident (read it all; understood it all reasonably well)


Reviewer 4

Summary

This paper improves the Fisher vector representations with a mixture of factor analyzers (MFA). The Fisher score with MFA (MFA-FS) is used as an image representation for holistic image classifiers and object-to-scene transfer. Overall, the proposed method is somehow novel and the experimental results show its effectiveness.

Qualitative Assessment

This paper argues the limitation of the standard GMM of diagonal covariance and derives a better model, MFA, to generate the Fisher score. The GMM-FV and MFA-FS are compared on the ImageNet based classifiers and object-to-scene transfer. Here are some comments: 1. In Table 3, the results of [1] based on Fisher Vector pooling of CNN performs better than MFA-FS. The authors need to provide more explanations and discussions. 2. Although achieving the state-of-the-art results, the contribution for the naïve combination of features from MFA-FS and Places CNN is limited as the other methods can easily run the similar procedure. More importantly, it is necessary for authors to provide more analysis on how object-based and scene-based representations help each other.

Confidence in this Review

2-Confident (read it all; understood it all reasonably well)


Reviewer 5

Summary

This paper studies Fisher Vector coding for deep CNN features for the application of task-based transfer, that is, utilizing the CNN features trained for object-level recognition to serve scene-level recognition. It first identifies the cause that GMM-based FV method does not work effectively for high-dimensional CNN features, and then proposes a new FV coding scheme based on the technique of mixture of factor analyzer (MFA). It is argued that MFA can better characterize the distribution of high-dimensional CNN features. Experimental study on benchmark data set demonstrates that the proposed method can effectively utilize object-level deep features to perform scene-level recognition.

Qualitative Assessment

The contribution of this work consists of two parts. First, it proposes the MFA-based FV coding method to pool object-level deep features. Although the MFA is a known technique, employing it for FV coding is novel and effective. Second, it shows that by using the proposed coding method, better scene-level recognition performance can be obtained by pooling the object-level features from image patches across multiple scales. Comparatively, the second contribution (built upon the first contribution) seems to be more interesting and significant. This paper is clear written and easy to follow. Especially, the design of experiments and the analysis of experimental results provide several important insights on this task-based transfer. They will be helpful for the research of scene-level recognition. Comments: 1. Does the effectiveness of MFA imply that the high-dimensional deep features are essentially generated by a lower-dimensional latent factor vector z? If yes, can this property be explained from or linked to the structure of the deep CNN (say, Alex or VGG) networks that generate the deep features? Please comment. 2. Line 183-190: The explanantion of the difference and advantage over existing work [10] and [11] can be made clearer. 3. In Table 3, MFA-FS (VGG) is still slightly lower than FV+FC(VGG)[1] (i.e., 81.43 vs. 81.7). This seems to be inconsistent with the expected advantage of MFA for FV coding. Please comment. Typo: Line 298: While they has (have) been... After the rebuttal: Thank the authors for the rebuttal. After reading the comments of other reviewers and the rebuttal, I think this is a nice piece of work with simple but insightful ideas that will attract extensive attention. I would like to maintain the original rating.

Confidence in this Review

2-Confident (read it all; understood it all reasonably well)


Reviewer 6

Summary

The paper claims that mixture of factor analyzers (MFA) is a better approximation compared to Fisher vector polling for transfer learning of high-order CNN features between object detection and scene classification tasks. This is supported by bunch of experiments that show improvements over other methods with considerable margins.

Qualitative Assessment

1. There is no theoretical justification of why MFA outperforms FV on transfer learning form object level to holistic scene descriptor. The main argument of the paper about "... inability of the standard GMM ... to provide good approximation ..." in L73-75 needs proof or reference to appropriate literature rather than only experiment results. 2. Authors claim "... MFA-FS captures highly discriminative covariance statistics ..." in L80-81, but I could not find any support of this except L161-162 which refers to full covariance S compared to Gaussian diagonal covariance. It needs to clarify why full covariance in MFA is the key to transfer learning problem on CNN features. I reckon it as a week argument although it was considered as second contribution of the paper because; any other dictionary learning method with full covariance should generate the same improvement as MFA according to authors' reasoning. 3. To my knowledge, section 2 of the paper seems a review of FV and MFA which does not clarify the novelty of authors' claimed contributions. An experienced reader is already aware of these formulations; hence it is expected to see the focus of formulation towards main claims which I could not see them there. 4. The paper seems like an experimental work which just plugs an idea to bunch of experiments to beat state-of-the-arts. I personally think that it may be of interest of computer vision community or venues like CVPR or WACV, but NIPS is a top-tier machine learning event which engineering tricks could not be that appreciated without sound theory and understanding in the background. 5. My other concern is about deployment of PCA with MFA-FS scores. Authors claim that MFA is employed due to its better ability to approximate nonlinear manifold of CNN. I expect that applying PCA to this high-order descriptor should degrade the accuracy much higher than what's reported in table 4, because PCA gets rid of many local sub-spaces with second and higher derivatives. Actually, it must diminish the advantage of MFA on factorizing to high-order sub-spaces, but the results remain almost the same i.e. 79.9 vs 79.3 on MIT. If the key contribution of the paper is a better description of high-dimensional and nonlinear CNN feature space by MFA, I’m wondering that it still remains competitive after a simple PCA.

Confidence in this Review

2-Confident (read it all; understood it all reasonably well)